# L³: Large Lookup Layers

**Albert Tseng** [1]   **Christopher De Sa** [1]

## Abstract

Modern sparse language models typically achieve sparsity through Mixture-of-Experts (MoE) layers, which dynamically route tokens to dense MLP "experts." However, dynamic hard routing has a number of drawbacks, such as potentially poor hardware efficiency and needing auxiliary losses for stable training. In contrast, the tokenizer embedding table, which is natively sparse, largely avoids these issues by selecting a single embedding per token at the cost of not having contextual information. In this work, we introduce the Large Lookup Layer (L³), which generalizes embedding tables to model decoder layers as a means of further scaling sparsity. L³ layers use static token-based routing to aggregate a *set* of learned embeddings per token in a *context-dependent way*, allowing the model to efficiently balance memory and compute by caching information in embeddings. L³ has two main components: (1) a systems-friendly architecture that allows for fast training and CPU-offloaded inference with no overhead, and (2) an information-theoretic embedding allocation algorithm that effectively balances speed and quality. We empirically test L³ by training transformers with up to 2.6B active parameters and find that L³ strongly outperforms both dense models and iso-sparse MoEs in both language modeling and downstream tasks.

## 1. Introduction

Recent advancements in language modeling have shown that increasing the sparsity (i.e. the ratio of total to active parameters, see Section 2) of a model can lead to significant improvements in downstream performance (Shazeer et al., 2017; Dai et al., 2024; Yu et al., 2025). The canonical way to do this is with a Mixtures-of-Experts (MoE) architecture

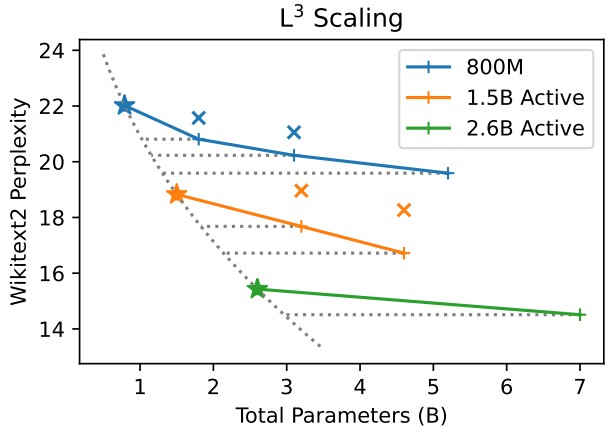

*Figure 1.* L³ parameter scaling. Adding sparsity with L³ layers ($+$) significantly improves performance over iso-FLOP dense models ($\star$) and iso-FLOP, iso-sparse MoEs ($\times$). See Appendix for FLOPs scaling.

(Shazeer et al., 2017), which replaces the MLP block of each decoder layer with a single router and multiple MLP "experts." In an MoE, the router performs context-dependent routing of each token to a subset of the experts; this subset is "activated" for that particular token.

While MoEs perform well, context-dependent routing is not very systems-friendly (Zoph et al., 2022). Routers must be trained to avoid collapse and poor load balancing, and since the exact experts a token activates are not known until the token reaches the router, experts cannot be offloaded without additional overhead (Wang et al., 2024). This raises the question: are there alternative sparse architectures that preserve the benefits of contextual routing without the systems baggage? Indeed, a partial solution already exists in modern language models: the tokenizer embedding table is an extremely sparse layer that has low systems overhead — but no contextual information.

Inspired by this, we introduce the Large Lookup Layer (L³), which generalizes the concept of an tokenizer embedding table to function within decoder layers in a *context-dependent* way. The core of L³ is a large collection of token-specific embeddings that act as a learned lookup table for the model to cache information in. We empirically show that this allows the model to shortcut computation around decoder

[1]Department of Computer Science, Cornell University. Correspondence to: Albert Tseng <albert@cs.cornell.edu>.

layers. $L^3$ is largely orthogonal to the MoE architecture; we propose $L^3$ as a way to further scale sparsity in models rather than a replacement for MoE layers.

The key to making $L^3$ fast is a systems-friendly architecture that uses static context-*independent* "routing" to select a fixed set of embeddings per token ID. Unlike an MoE, the exact $L^3$ parameters needed are known the moment a token is generated. This allows us to offload $L^3$ parameters and overlap fetching with pre-$L^3$ computation, something not traditionally possible with context-dependent routing. To ensure that these embeddings are used effectively, the embeddings are attended to by the token hidden state, allowing the model to perform a context-*dependent* "lookup."

The main tuning knob of quality in $L^3$ is how embeddings are allocated to token IDs. We show that by viewing $L^3$'s static token router as a replacement for a context-dependent router, we can use lossless compression algorithms such as LZW (Welch, 1984) to determine embedding allocations by the frequency of codewords. Our LZW-based allocation algorithm allows us to effectively double the perplexity gap of $L^3$ over a dense model versus using a naive uniform embedding allocation.

The rest of this paper is organized as follows. First, we introduce the $L^3$ layer and our token-embedding allocation algorithm. Then, we describe strategies for fast training and inference. Finally, we empirically evaluate $L^3$ by pretraining models with up to 2.6B active parameters and show that $L^3$ outperforms iso-FLOP dense models and iso-FLOP, depth, and sparse MoEs, all without the systems issues of context-dependent routing. In summary, we:

1. Introduce the Large Lookup Layer ($L^3$), which generalizes sparse embedding tables by using contextual information to aggregate a learned set of embeddings for each token ID.

2. Introduce information-theoretic embedding allocation algorithms for token-embedding mappings in $L^3$.

3. Pretrain models with up to 2.6B active parameters and empirically show that $L^3$s outperform iso-FLOP dense models and iso-FLOP, depth, and sparse MoEs.

4. Show that $L^3$s are hardware-friendly, supporting fast training, inference, and offloading.

## 2. Background and Related Work

It is important to differentiate between two notions of sparsity in the wider literature. In the classic setting, a sparse model such as a 2:4 model is one that uses sparse weight matrices (only a fraction of the weights are non-zero) with a fixed sparsity pattern (Lu et al., 2023; Frantar & Alistarh, 2023). A more recent notion, which we use in this paper,

considers sparse models as those with more total parameters than active parameters (only a fraction of the weights are used for each input). In this sense, a classical sparse model would *not* be sparse, since all parameters are active for all inputs. A unifying property of both notions of sparsity is that sparsity patterns generally have to be *structured* to be hardware-efficient (Gray et al., 2017). Below, we describe approaches in both architectural and parameter sparsity.

### 2.1. Architectural Sparsity

The classical notion of a sparse model follows the definition of sparsity in a matrix. In such models, certain parameters are set to 0 and can be skipped during computation. One approach to achieving hardware-friendly architectural sparsity is to use $n : m$ sparse patterns (e.g. 2-of-4) to accelerate GEMMs (Lu et al., 2023; Choquette, 2023). In contrast to quantization, where each weight is retained and compressed to use fewer bits (Frantar et al., 2023; Chee et al., 2024; Tseng et al., 2024; 2025a;b;c), $n : m$ models set $m - n$ weights to 0 per block of $m$ weights. The main difficulty in obtaining a good $n : m$ model is in selecting the nondifferentiable sparsity mask. However, once a mask is chosen, the resulting weights can be "retrained" to minimize divergence from the original model. This also allows a $n : m$ sparse model to be pretrained from scratch with a fixed sparsity mask, although we are not aware of any state of the art models that do this.

### 2.2. Mixtures-of-Experts

One significant issue with fixed-pattern sparse models is that there are limited empirical benefits from using them. The Mixtures-of-Experts (MoE) architecture solves this by instead using context-dependent routing to *dense* experts in the MLP layers of a model. In an MoE, the MLP layers of a regular dense model are replaced by a router and set of experts. Each token is routed to a subset of all the experts (e.g. 2 out of 32 experts) based on its hidden state representation, meaning that the entire context is used to route the token. Unlike in a classical sparse model, a different set of parameters is used per input in an MoE, so there *is* a significant benefit from using sparsity.

The performance of an MoE is highly dependent on its routers' abilities to sort tokens into different experts (Zoph et al., 2022). If no meaningful separation occurs or all tokens collapse onto a single router, then an MoE layer becomes a dense layer. In a vanilla router, the output of a token is computed as $y = \sum_{e \in E_x^*} r(x, e)e(x)$, where $r(x, e)$ is the router score for $x$ and $e$, $e(x)$ is the expert output, and $E_x^*$ is the set of active experts for $x$ (Muennighoff et al., 2025). $r(x, e)$ is typically implemented so that $\sum_{e \in E_x} r(x, e) = 1$, where $E_x$ is the set of *all* experts. To prevent collapse, algorithms such as top-$k$ and expert-choice routing as well

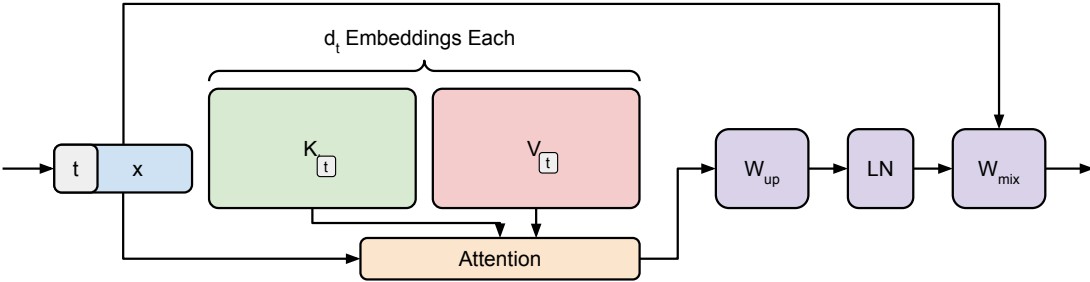

*Figure 2.* The basic architecture of a L³ layer for a single token. The token ID $t$ is used to select embeddings $K_t, V_t$. Embeddings are aggregated with attention from the token hidden state $x$. The attention output is combined with the residual stream (hidden state $x$) to produce the L³ output. Embeddings are assigned to token IDs with an information-theoretic LZW-based algorithm.

as auxiliary losses such as load-balancing and "router z" losses have been proposed to stabilize training (Shazeer et al., 2017; Dai et al., 2024; Muennighoff et al., 2025; Zoph et al., 2022; Zhou et al., 2022).

Finally, although MoEs usually use dense MLPs as experts, MoEs can still be difficult to run at scale. Large MoEs have trillions of parameters that require multiple devices to hold and careful sharding to minimize overhead (Guo et al., 2025). A poorly balanced MoE with overloaded experts can also cause low device utilization, and dropping tokens can cause poor model quality (Muennighoff et al., 2025). Furthermore, since the exact experts a token will hit are unknown until an MoE layer is reached, experts cannot easily be offloaded without additional overhead. A large body of literature has developed approaches toward accelerating MoE training and inference, which we do not detail for brevity (Guo et al., 2025; Aimuyo et al., 2025).

### 2.3. Embedding Layers

While MoEs are the predominant way to achieve parameter sparsity in models, the tokenizer embedding table in every language model is also a form of parameter sparsity. In it, each token "activates" a single row like a lookup table to obtain the initial embedding for modeling. Although these parameters are computationally "free," they are crucial to modeling, suggesting that it may be beneficial to generalize embedding tables beyond tokenizers. Indeed, prior works have used this idea to improve model quality.

One of the earliest forms of this idea was the Product Key Network (PKN) (Lample et al., 2019; Huang et al., 2025). Like L³, PKNs select keys in a large embedding table to aggregate into the model residual stream. However, PKNs use a query vector formed from the current hidden state. This means that they effectively perform context-based routing like an MoE, which makes L³'s systems benefits like offloaded inference impossible.

In SCONE, Yu et al. (2025) collect a set of "f-grams" that each get a single embedding in a large embedding table

at the *beginning* of the model. During training, a small transformer produces embeddings for these f-grams, and the embedding corresponding to an input token is the embedding of the longest f-gram that is a suffix of the prefix ending at that token. During inference, the f-gram embedding table is cached. SCONE essentially expands the tokenizer embedding table to a local context embedding table, allowing the model to "skip computation" on cached f-grams.

In contrast, the recent Engrams work places embedding tables *throughout* the model and uses local suffix information to select a set of embeddings per token (Cheng et al., 2026). Similar to our approach, these embeddings are aggregated based on the current hidden state with full context information. However, unlike L³, Engrams contains many additional steps such as n-gram hashing, tokenizer compression, and pooling. Our experiments suggest that L³ scales similarly to Engrams and that the key to both approaches is in the large embedding table. Since Engrams is concurrent work, we leave a full comparison for the future.

Finally, works such as Cartridges (Eyuboglu et al., 2025) have targeted learning a *KV cache*, which is in some sense conceptually aligned with how L³ attends to learned embedding layers. These works target retrieval tasks where multiple prompts attend to a shared context. In Cartridges, Eyuboglu et al. (2025) showed that it was possible to learn a heavily compressed coreset of KVs per task, and that the learned KVs could be swapped across tasks. We suspect that "swapping" L³ layers across tasks is also possible, but we leave this for future work.

## 3. Large Lookup Layers

Here, we describe the L³ architecture and key design choices that affect its performance. At the level of a single token, an L³ layer takes as input a hidden state $x \in \mathbb{R}^{d_{in}}$ and corresponding token ID $t \in \{1, \ldots, |\tau|\}$, where $|\tau|$ is the vocab size of the tokenizer $\tau$. The L³ layer is parameterized by a mixing matrix $W_{mix} \in \mathbb{R}^{d_{out} \times (d_{in} + d_{up})}$, an "up projection" matrix $W_{up} \in \mathbb{R}^{d_{up} \times d_{emb}}$, a LayerNorm (Zhang & Sennrich,

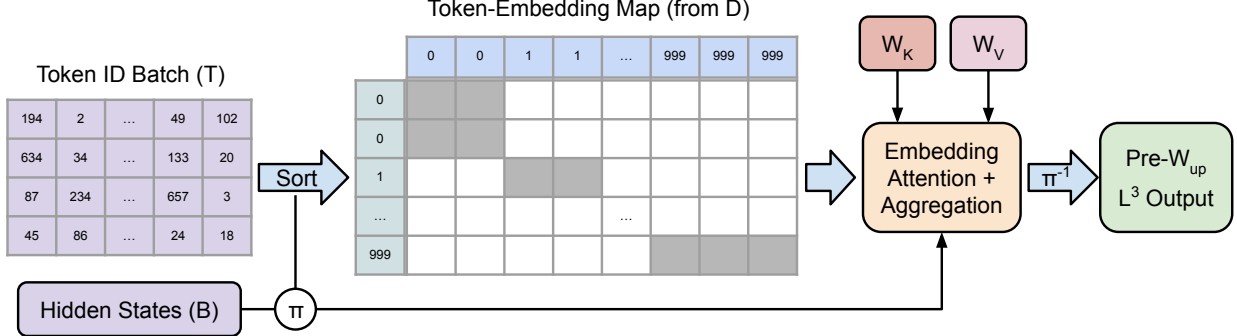

**Figure 3.** Efficient training with $L^3$. Since $L^3$ only performs channel mixing, all tokens in a batch can be sorted to form a block-diagonal "attention mask" for $L^3$ embedding aggregation. This allows $L^3$ to use fast kernels such as MegaBlocks or FlexAttention. The only overhead is sorting tokens, which can be done ahead of time, and inverting the sort, which is trivial.

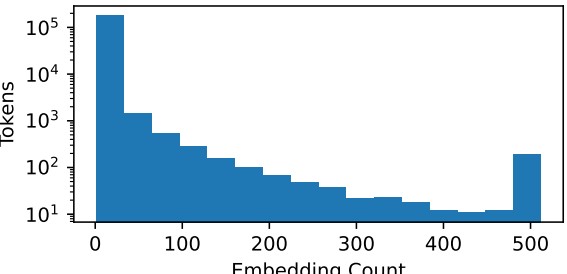

**Figure 4.** Fast offloaded inference with $L^3$. Since the set of needed $L^3$ parameters is known when a token is generated, $L^3$ parameters can be offloaded and prefetched during pre-$L^3$ compute.

**Figure 5.** Histogram of embedding count for 710K embeddings, a 180K vocab BPE tokenizer, and Algorithm 1 with $k = 512$. "Common" tokens such as "then" get 512 embeddings while "rare" tokens such as "orm" get 1 embedding.

2019), and a sequence of "key" and "value" matrices $K = \{K_1, \ldots, K_{|\tau|}\}$ and $V = \{V_1, \ldots, V_{|\tau|}\}$, where matrix $K_t$ is of shape $(d_t, d_{\text{in}})$ and matrix $V_t$ is of shape $(d_t, d_{\text{emb}})$, for a sequence of sizes $D = \{d_1, \ldots, d_{|\tau|}\} \in \mathbb{Z}^+$. These sizes must be selected as hyperparameters before training via some embedding *allocation* procedure. The $L^3$ layer then outputs

$$
\begin{aligned}
&L^3(x, t; W_{\text{mix}}, W_{\text{up}}, K, V) \in \mathbb{R}^{d_{\text{out}}} \\
&= W_{\text{mix}} \begin{bmatrix} \text{LayerNorm}(W_{\text{up}}(V_t^T \text{Softmax}(K_t x))) \\ x \end{bmatrix}.
\end{aligned}
$$

All parameters are learned except for $D$, which is the main tuning knob for quality and speed in $L^3$.

### 3.1. Embedding Allocation

For a fixed budget of $v = \sum_i d_i$ embeddings, different allocations of embeddings to tokens can significantly affect the performance of a $L^3$ layer. We wish to emulate the behavior of a *context-dependent* router (like in an MoE) with $D$. A natural lookup-based approximation to such a router is to cache commonly occurring suffixes of contexts and to match the longest suffix of a given context when selecting an embedding. This approximation reduces the problem of building an allocation over $v$ to finding a representative set of suffixes for a given corpus. This is effectively a dual problem to lossless text compression, where the goal

is to find a set of codes (tuples of tokens) that minimizes the expected coded length of an input sequence under a contiguous longest-code matching (Ziv & Lempel, 1978). As such, we can use a classic compression algorithm to build a set of codewords with which to allocate embeddings.

In particular, we introduce Algorithm 1, a variant of the LZW compression algorithm (Welch, 1984). In Algorithm 1, we first build a set of codewords and frequencies by scanning a corpus and finding the longest suffix that has not been seen yet. When a new suffix is obtained, the frequency of that suffix is set to 1 and the frequency of the immediate suffix of *the new suffix* is increased by 1. Then, we iterate over codewords in descending frequency order and allocate an embedding to token $t$ if the codeword ends in $t$. We do this until we hit a target number of total embeddings, with the additional condition that each token gets at least one codeword and no more than $k$. These caps ensure that each token is represented and bounds the worst case number of active parameters for any token, which determines the FLOPs and on-device memory a $L^3$ layer needs.

Figure 5 shows the result of running Algorithm 1 on the

**Algorithm 1** LZW Embedding Allocation

---

**input** Dataset $\mathcal{D}$, Tokenizer $\tau$, Target Embedding Count $v$,
   Max Per-Token Embedding Count $k$.
   Dictionary $C \leftarrow \{\}$
   **for** Byte Sequence $s \in \mathcal{D}$ **do**
      $s_\tau \leftarrow \tau(s)$
      **for** $0 \leq i < |s_\tau|$ **do**
         $j \leftarrow 0$
         **while** $s_\tau[i-j:i+1] \in C$ **do**
            $j \leftarrow j+1$
         **end while**
         $C[s_\tau[i-j+1:i+1]] \leftarrow C[s_\tau[i-j+1:i+1]]+1$
         $C[s_\tau[i-j:i+1]] \leftarrow 1$
      **end for**
   **end for**
   Convert $C$ to list of keys sorted descending by value.
   Initial allocation $A = \{1, 2, 3, ..., |\tau|\}$, Count $A_C = \mathbb{1}^{|\tau|}$
   $i \leftarrow 0$
   **while** $|A| < v$ **do**
      **if** $A_C[C[i][-1]] < k$ **then**
         $A \leftarrow \begin{bmatrix} A & C[i][-1] \end{bmatrix}$
         $A_C[C[i][-1]] \leftarrow A_C[C[i][-1]] + 1$
      **end if**
      $i \leftarrow i+1$
   **end while**
**output** $\text{sort}(A)$

---

tokenizer we use for our experiments. The resulting allocation distribution is slightly heavy tailed and close to Zipf's law (Piantadosi, 2014). Although we use Algorithm 1 in our experiments due to its strong performance over uniform allocation (Figure 7(C)), we emphasize that the basic $\text{L}^3$ architecture allows user choice in determining the best allocation for a given task as long as the maximum number of embeddings per token is relatively low ($\leq 10^3$).

### 3.2. Efficient Training and Inference

To actually implement $\text{L}^3$, we concatenate $K$ into a single matrix $W_K \in \mathbb{R}^{v \times d_{\text{in}}}$ and likewise $V \to W_V \in \mathbb{R}^{v \times d_{\text{emb}}}$. Due to $\text{L}^3$'s static routing, the active parameters in $W_K$ and $W_V$ are known the moment a token or sequence is generated. We can take advantage of this in both training and inference. First, we can offload $\text{L}^3$ parameters with minimal overhead. During training, we empirically observe that for our 2.6B parameter model and a context length of 2048, only $\approx 100\text{M}$ $\text{L}^3$ parameters are active per microbatch of 8K tokens. In small batch inference, the number of required parameters is even smaller. These parameters can be prefetched during computation *before* $\text{L}^3$ layers are reached (Figure 4). This offloading also makes context parallel style sharding (Liu et al., 2024) straightforward, since offloaded $\text{L}^3$ add no more complexity for context parallelism than a dense MLP.

Second, in large-batch settings, we can sort the entire batch for better memory access patterns (Figure 3). Consider an input batch of hidden states $B = \{x_1, ..., x_n\}$ with corresponding tokens $T = \{t_1, ..., t_n\}$. Running $\text{L}^3$ on $B$ is equal to running regular attention with $Q = B$, $K = W_K$, and $V = W_V$ with an attention mask equal to the allowed tokens from $T$. Since $\text{L}^3$ only performs channel mixing, we can sort $B$ by $T$ so the resulting "attention mask" is block-diagonal. This allows us to either use off-the-shelf attention kernels like FlexAttention (Dong et al., 2024) or simply loop over diagonal blocks. The only additional overhead is permuting the incoming hidden states to the $\text{L}^3$ layer and inverting, which is negligible.

In contrast, these optimizations are difficult to implement in MoEs. In an MoE, the active experts per token are not known until the relevant router is reached. Furthermore, since the sparsity rate is much lower in a single MoE layer ($4 - 32$) than a $\text{L}^3$ layer ($> 1000$), a single training batch is almost guaranteed to hit *all* experts in a model, whereas this is *impossible* in $\text{L}^3$ since $|\tau| \gg$ any reasonable batch size. For MoEs, this prevents offloading, which increases the number of devices an MoE must be sharded over and adds overhead to parallelism strategies.

## 4. Experiments

To test $\text{L}^3$, we pretrain Llama-based transformers with $\text{L}^3$ layers added between decoder layers (Grattafiori et al., 2024). The goal of our experiments was to test the effect of adding embeddings through $\text{L}^3$ layers, so we do not replace the MLP layers of a model with $\text{L}^3$ layers like an MoE does. It is also important to emphasize that $\text{L}^3$ is not a replacement for the traditional MoE architecture, as it targets different modeling behaviors and *per-layer* sparsity rates. However, since MoEs are the predominant sparse architecture, we do compare to iso-sparse MoEs and show that $\text{L}^3$ is generally more efficient. We leave the problem of using both MoEs and $\text{L}^3$ in the same model for future work.

Our main experiments use 3 model sizes: 800M (400M decoder), 1.5B (1B), and 2.6B (1.9B) active parameters, which we train for $\approx 10\text{B}$, 20B, and 30B tokens of FineWeb-Edu (Penedo et al., 2024), respectively, with a context length of 2048. We use a standard BPE tokenizer (Gage, 1994) with 180000 tokens that achieves 4.7 bytes per token on FineWeb-Edu. We target sparsity ratios of 2-4$\times$ the active parameter count, which are usually achieved with 1-2 $\text{L}^3$ layers across the entire model. Unless otherwise specified, we use a total embedding table size of $v = 710000$ and cap the number of embeddings per token to $k = 512$. Exact model dimensions are in the Appendix.

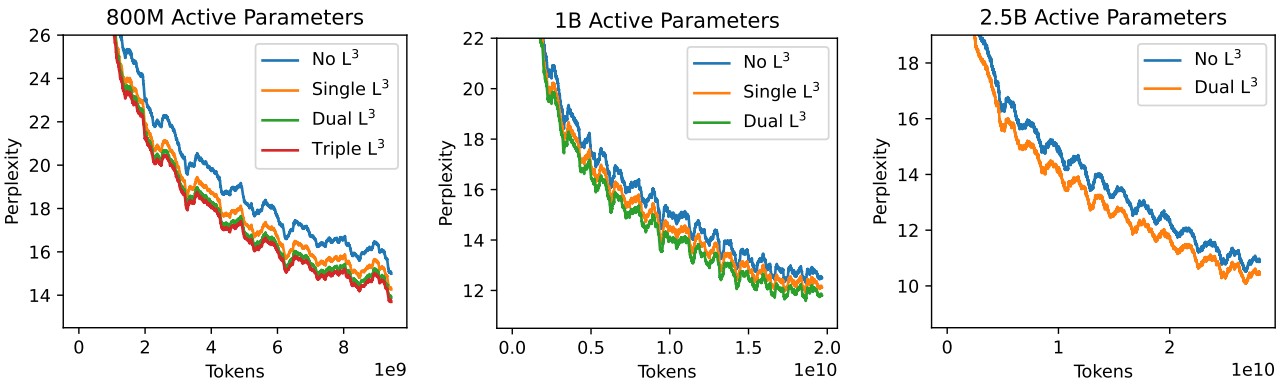

*Figure 6.* Training perplexity on FineWeb-Edu, dense and L$^3$ transformers. L$^3$ layers significantly improve performance.

*Table 1.* Downstream performance. Adding L$^3$ layers significantly improves performance over iso-FLOP dense models. Active parameters / token is calculated with an empirical expectation over Fineweb-Edu. *Uses a wider L$^3$ layer than other 800M-class experiments. Downstream performance of MoE equivalents can be found in the Appendix. **Best in bold**, second best underlined.

| ACTIVE PARAMS / TOK | L$^3$ LAYERS | TOTAL PARAMS | PPL. WIKI2 | 0-SHOT | | | | | |
|---|---|---|---|---|---|---|---|---|---|
| | | | | AVG | ARCC | ARCE | HELLASWAG | PIQA | WINOGRANDE |
| 809M | 0 | 809M | 22.02 | 48.28 | 27.56 | 60.52 | 34.32 | 67.13 | 51.85 |
| 795M | 1 | 1.8B | 20.81 | 49.10 | 28.21 | 62.03 | 35.58 | 67.52 | 52.17 |
| 803M | 2 | 3.1B | 20.23 | 49.45 | 28.67 | 62.92 | 36.09 | 67.83 | 51.72 |
| 818M | 3* | 5.2B | **19.59** | **50.25** | **29.01** | **63.64** | **36.32** | **68.23** | **54.06** |
| 1.5B | 0 | 1.5B | 18.83 | 51.93 | 29.18 | 66.93 | 40.23 | 67.96 | 55.37 |
| 1.5B | 1 | 3.1B | 17.68 | 53.09 | 31.91 | 67.34 | 41.04 | 69.37 | 55.81 |
| 1.5B | 2 | 4.6B | **16.72** | **53.84** | **33.11** | **67.63** | **41.93** | **70.67** | **55.88** |
| 2.6B | 0 | 2.6B | 15.43 | 55.59 | 36.35 | 68.86 | 44.20 | 71.22 | 57.30 |
| 2.6B | 2 | 7B | **14.51** | **56.98** | **38.21** | **71.21** | **44.95** | **71.71** | **58.80** |

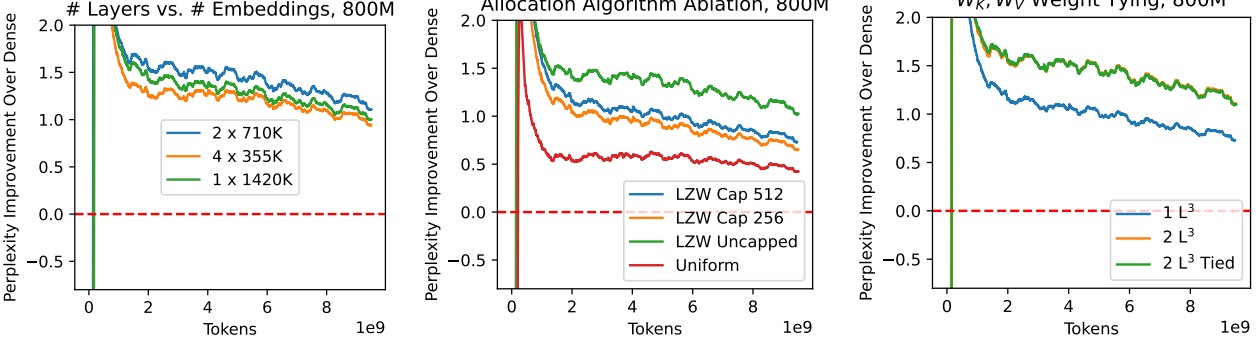

*Figure 7.* (L) Ablation on the number of L$^3$ layers and parameters per L$^3$ layer for a target sparsity rate. (C) Ablations on LZW allocation. Uncapped LZW performs the best but has poor access guarantees, with the most "common" token getting over 20K embeddings out of 710K. Using uniform assignment across tokens performs poorly. (R) Ablations on weight tying $W_K$ and $W_V$. Weight tying has essentially no effect on L$^3$'s performance while almost halving the sparsity rate and data transfer volume.

### 4.1. Pretraining Experiments

Figure 6 and Table 1 show the effect of L$^3$ layers in iso-FLOP settings. At all sizes, adding L$^3$ layers significantly improves both training perplexity and downstream performance. This improvement is consistent throughout the entire training run, unlike in an MoE where learning a discrete router can make the model initially worse than a dense baseline. Figure 1 shows the effect of scaling sparsity with L$^3$. Using a power law $P = aN^b + c$ to fit perplexity $P$ against parameters $N$ for dense models, we find that increasing the sparsity linearly with L$^3$ layers results in a roughly linear increase in parameters for an equivalent dense model. Futhermore, we find that increasing sparsity for a fixed

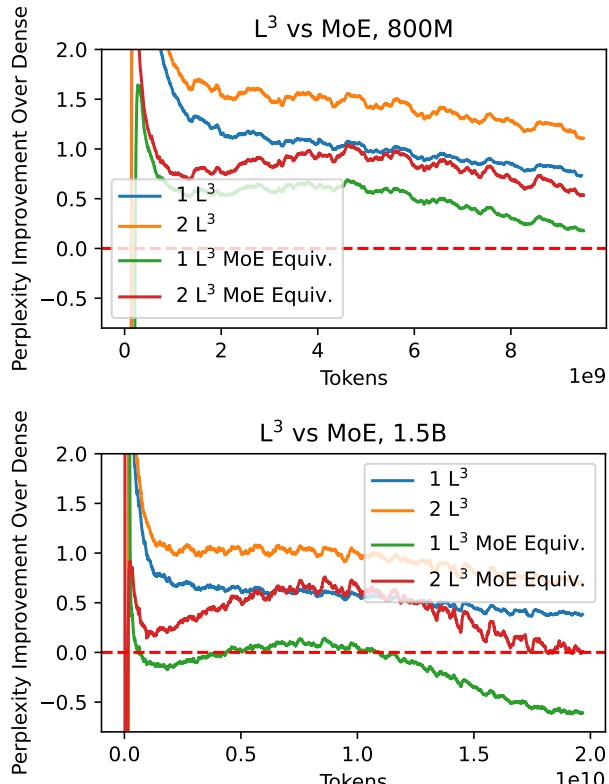

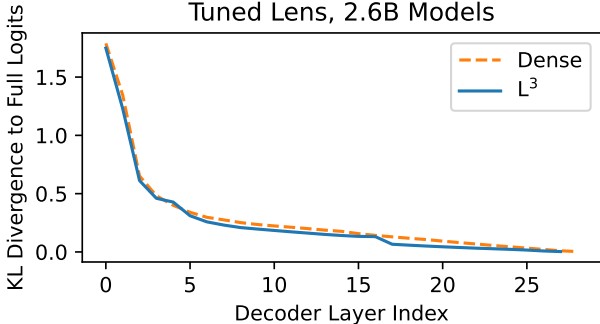

Figure 10. Tuned lens on 2.6B $L^3$ and dense models. $L^3$ layers (placed after layers 4 and 16) induce sharp drops in KL across decoder layers, indicating that the model is caching information in $L^3$ layers.

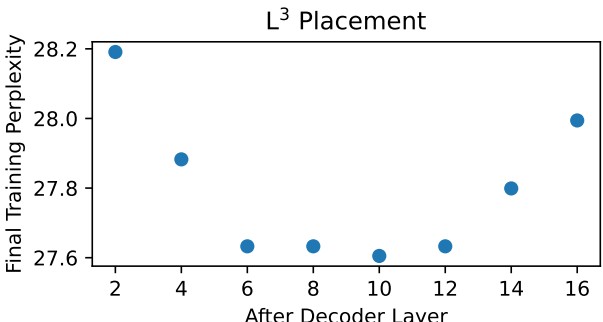

Figure 8. Perplexity gap vs. dense model for $L^3$ models and iso-FLOP and iso-sparse MoEs. At all tested sparsity levels, $L^3$s outperform equivalent MoEs.

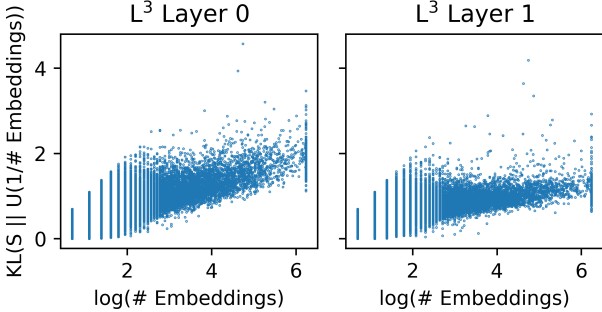

Figure 11. Average KL divergence between $L^3$ score distribution and uniform distribution over allowable embeddings vs. entropy of uniform distribution from the 2.6B $L^3$ model. Each point is a token index. The $x$-axis corresponds to the log of the number of allowable embeddings for that token index.

Figure 9. Final training perplexity of a 800M class model with a single $L^3$ layer with 355K embeddings placed after different decoder layers. Models trained for 1B tokens.

active parameter count improves performance, but not as much as increasing the active parameter count (of course, this would increase FLOPs). These observations mirror the empirical scaling behavior of MoEs (Krajewski et al., 2024), without context-dependent routing.

Figure 8 compares the 800M and 1.5B class $L^3$ models against iso-FLOP, depth, and sparse MoE models. We follow OLMoE recommendations when training MoEs and

use a standard softmax-based router with a load balancing loss and router z-loss (Muennighoff et al., 2025; Zoph et al., 2022). We target around 20 total experts per layer and adjust the expert intermediate dimension to hit a specific total sparsity. Exact details are in the Appendix. At all tested sparsity rates, $L^3$ models maintain a larger perplexity gap over the baseline dense model throughout training. Notably, at the 800M parameter scale, an MoE equivalent to two $L^3$ layers underperforms a single $L^3$ layer, and at the 1.5B scale, an MoE equivalent to one $L^3$ layer underperforms the *dense* model. These results suggest that while MoEs may require certain token budgets or sparsity rates to significantly outperform dense models due to learnable routing, the benefits of $L^3$ layers are immediately apparent.

### 4.2. Architectural Design Choices

The basic $L^3$ layer allows for certain implementation choices, such as varying the number of embeddings per $L^3$ layer and where to place layers. Figure 7(L) shows an ablation between using 2 $L^3$ layers with 710K embeddings

each (the $L^3$ size used in all previous experiments), 4 layers with 355K embeddings each, and 1 layer with 1420K embeddings. Although all 3 setups perform similarly, using one very large layer of 1420K embeddings limits the amount of time available to prefetch parameters when using offloading. On the other hand, using multiple smaller layers restricts where in the model $L^3$ layers can be added. To better understand the effect of this, Figure 9 ablates the effect of placing a single $L^3$ layer after a target decoder layer. Placing layers too early limits how much context information can be used to aggregate embeddings, and placing layers too late limits how much impact the embeddings have.

Figure 7(C) shows an ablation on the cap $k$ used in our LZW allocation algorithm (Algorithm 1). Using LZW allocation with $k = \infty$ results in the best performance, but gives poor worst case guarantees. The token with the most embeddings consumes $> 20K$ out of 710K embeddings, meaning that O(50M) parameters need to be loaded if that particular token is hit. Capping the number of embeddings at $k = 512$ limits the worst case number of parameters needed to O(1M) while still offering good performance, and capping at 256 continues this trend. In contrast, assigning an equal number of embeddings to each token (4 per token for 720K total) performs poorly, indicating that Algorithm 1 is crucial to maximizing $L^3$'s performance.

Finally, Figure 7(R) shows an experiment tying $W_K$ and $W_V$ together. Although we did not use weight tying in our main experiments to better analyze the behavior of $L^3$, weight tying has essentially no effect on the quality of $L^3$ while almost halving the effective sparsity (i.e. better quality at the same sparsity rate) and data transfer volume (lower inference latency).

### 4.3. What are $L^3$ Layers Doing?

Intuitively, $L^3$ layers store information that the model associates with certain token IDs. Figure 10 shows the KL divergence of the output after each decoder layer with respect to the final model output under a tuned lens (Belrose et al. (2025), see Appendix) that is trained to minimize the KL to the final model output. The $L^3$ curve exhibits two sharp drops in KL after layers 4 and 16, corresponding to where the 2 $L^3$ layers are placed in the model. In contrast, the dense model has a smooth decrease in KL across model depth, suggesting that $L^3$ layers cache information that would otherwise require decoder layers to recompute.

A natural question to ask is how $L^3$ layers aggregate cached information. Figure 11 plots the KL divergence between the $L^3$ score $\text{Softmax}(K_t x)$ and a uniform distribution over the allowed embeddings against the number of allowed embeddings. In both $L^3$ layers in the model (after decoder layers 4 and 16), the KL slowly increases with number of embeddings, suggesting that LZW allocation does a good

*Table 2.* Inference speed at various batch sizes for 2.6B active parameter models on one B200 GPU. The dense model has 2.6B total parameters and the $L^3$ model uses 2 CPU-offloaded $L^3$ layers with $k = 512, v = 710000$, giving 7B total parameters. Due to $L^3$'s static routing, CPU-offloading $L^3$ layers adds minimal inference overhead. All models use BF16.

| FIRST $L^3$ LAYER | BS=1 TOKS/S/SEQ | BS=8 TOKS/S/SEQ | BS=300 TOKS/S/SEQ |
|---|---|---|---|
| DENSE | 776 | 492 | 487 |
| 1 | 692 | 424 | 396 |
| 2 | 711 | 440 | 421 |
| 3 | 743 | 459 | 450 |
| 4 | 768 | 476 | 468 |
| 5 | 772 | 487 | 484 |

job of balancing access patterns across token frequency. The first $L^3$ layer also generally has a higher KL than the second, suggesting that the earlier $L^3$ layer is closer to a lookup layer whereas the second layer tends more towards aggregating across embeddings.

### 4.4. Training and Inference

Unlike the dynamic routing of an MoE, $L^3$'s static routing allows for relatively simple high-throughput training and inference. Our training setup uses a straightforward PyTorch implementation that iterates over active block diagonals in the "attention mask." This is sufficient to achieve 135K toks/s for an 800M parameter model on an 8×A100 node, or roughly 87% the 155K toks/s we achieved for an equivalent dense model. We view this number as an upper bound on the penalty for training a $L^3$ model, since the overhead of $L^3$ layer decreases with model size and using an optimized kernel should significantly improve throughput.

Table 2 shows a comparison of batch size 1 inference speed for various 2.6B class models on a single A100 SXM 80GB GPU ($\approx$ 2 TB/s Mem. BW). $L^3$ adds minimal overhead, regardless of whether its parameters are offloaded or not. This is possible for two reasons. First, unlike an MoE $L^3$'s routing depends only on the current token generated, meaning that all relevant $L^3$ parameters for the current token can be prefetched the moment a token is generated (Figure 4). Second, since we choose $k$ to be small enough that the total data movement for all $L^3$ parameters in a model is O(1MB) at the 2.6B scale, data transfers can easily be masked by compute in pre-$L^3$ layers. In the "worst case" scenario where the first $L^3$ layer happens after one decoder layer, $L^3$ only adds 10% overhead, and only 4 decoder layers are sufficient to fully mask any PCIe latency.

## 5. Conclusion

In this work, we introduce the $L^3$ layer, which generalizes the sparse tokenizer embedding table to process contextual

information in model decoder layers. $L^3$ layers contain two main components: a set of large embedding tables that are statically routed to based on token ID, and an information-theoretic token-embedding allocation algorithm. $L^3$ uses the incoming hidden state to attend to these embeddings, allowing the model to perform context-dependent aggregation without the overhead of context-dependent routing. We empirically show that $L^3$ significantly outperforms iso-FLOP dense models and iso-FLOP, depth, and sparse MoEs while adding minimal training and inference overhead, even when *CPU offloaded*. These results suggest that $L^3$ offers a new, more efficient axis of sparse scaling in models.

## Impact Statement

This paper presents work whose goal is to advance the field of Machine Learning. There are many potential societal consequences of our work, none which we feel must be specifically highlighted here.

## Acknowledgements

AT was supported by the NSF Graduate Research Fellowship. CD was supported by DARPA YFA D24AP00259 and NSF Career Award 2046760. We thank Together AI for compute resources, Edgar Marroquin for discussions, and Tao Yu for feedback on the idea and manuscript.

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

# A. Appendix

## A.1. Additional Results

*Table 3.* Downstream Results for MoE equivalent models. At all sparsity rates and model sizes, L$^3$ outperforms MoE models.

| Model | Avg | ArcC | ArcE | HSwag | PiQA | WinoG |
|---|---|---|---|---|---|---|
| 800M 1 L3 MoE equiv. | 48.18 | 27.25 | 60.27 | 34.79 | 67.36 | 51.22 |
| 800M 2 L3 MoE equiv. | 48.59 | 27.60 | 60.48 | 35.28 | 68.39 | 51.22 |
| 1.5B 1 L3 MoE equiv. | 51.68 | 30.63 | 65.65 | 38.89 | 70.18 | 53.04 |
| 1.5B 2 L3 MoE equiv. | 52.83 | 31.72 | 66.54 | 41.23 | 70.52 | 54.13 |

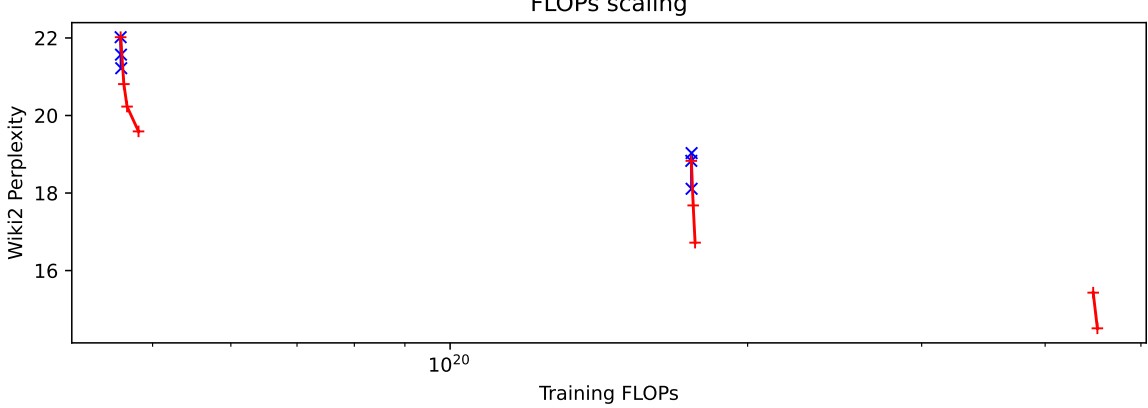

*Figure 12.* FLOPs scaling of L$^3$ ($+$) and MoE ($\times$) models.

## A.2. Model Architecture and Training Details

Our models use the standard Llama 3 decoder layer architecture implemented in HuggingFace Transformers (Wolf et al., 2020) as a basis. We use the Llama RMS Norm with $\epsilon = 10^{-5}$, and the RoPE scaling and $\theta$ configuration from Llama 3.2 1B. We train models with the AdamW optimizer (Loshchilov & Hutter, 2019) with the default PyTorch $\epsilon = 10^{-8}$ and a cosine schedule learning rate (Loshchilov & Hutter, 2017). We use the PyTorch implementations of FSDP sharding with BF16 mixed precision training and activation checkpointing (Paszke et al., 2019). We use a standard Softmax MoE router implementation adapted from the GPT-OSS HuggingFace implementation (OpenAI et al., 2025). We stream the FineWeb-Edu dataset from HuggingFace using the "train" split of the "Sample-350BT" slice using a shuffle buffer of size 10000 per GPU. We use one $8\times$A100 80G SXM4 node for each experiment, except for the 2.6B training runs, where we use 2 nodes per run. All plots use an EMA of 0.9 for visual clarity.

*Table 4.* Dense Model Configurations

| Size | 800M | 1.5B | 2.6B |
|---|---|---|---|
| Decoder Layers | 21 | 24 | 28 |
| Hidden Size | 1024 | 1536 | 2048 |
| Attention Heads | 32 | 32 | 32 |
| Attention Head Dim | 64 | 64 | 64 |
| Context Length | 2048 | 2048 | 2048 |
| Batch Size (Tokens) | 131072 | 131072 | 131072 |
| Peak Learning Rate | $3 \times 10^{-4}$ | $3 \times 10^{-4}$ | $3 \times 10^{-4}$ |
| Min Learning Rate | $3 \times 10^{-5}$ | $3 \times 10^{-5}$ | $3 \times 10^{-5}$ |
| Weight Decay | 0.1 | 0.1 | 0.1 |
| Gradient Clipping | 1 | 1 | 1 |
| Intermediate Dim | 4096 | 6144 | 8192 |
| Tokens | $10^{10}$ | $2 \times 10^{10}$ | $3 \times 10^{10}$ |
| Tokenizer Size | 180000 | 180000 | 180000 |

*Table 5.* $L^3$ Model Configurations

| Size | 800M 1 $L^3$ | 800M 2 $L^3$ | 800M 3 $L^3$ | 1.5B 1 $L^3$ | 1.5B 2 $L^3$ | 2.6B 2 $L^3$ |
|---|---|---|---|---|---|---|
| Decoder Layers | 20 | 20 | 20 | 24 | 24 | 28 |
| Hidden Size | 1024 | 1024 | 1024 | 1536 | 1536 | 2048 |
| Attention Heads | 32 | 32 | 32 | 32 | 32 | 32 |
| Attention Head Dim | 64 | 64 | 64 | 64 | 64 | 64 |
| Context Length | 2048 | 2048 | 2048 | 2048 | 2048 | 2048 |
| Batch Size (Tokens) | 131072 | 131072 | 131072 | 131072 | 131072 | 131072 |
| Peak Learning Rate | $3 \times 10^{-4}$ | $3 \times 10^{-4}$ | $3 \times 10^{-4}$ | $3 \times 10^{-4}$ | $3 \times 10^{-4}$ | $3 \times 10^{-4}$ |
| Min Learning Rate | $3 \times 10^{-5}$ | $3 \times 10^{-5}$ | $3 \times 10^{-5}$ | $3 \times 10^{-5}$ | $3 \times 10^{-5}$ | $3 \times 10^{-5}$ |
| Weight Decay | 0.1 | 0.1 | 0.1 | 0.1 | 0.1 | 0.1 |
| Gradient Clipping | 1 | 1 | 1 | 1 | 1 | 1 |
| Intermediate Dim | 4096 | 4096 | 4096 | 6144 | 6144 | 8192 |
| Tokens | $10^{10}$ | $10^{10}$ | $10^{10}$ | $2 \times 10^{10}$ | $2 \times 10^{10}$ | $3 \times 10^{10}$ |
| Tokenizer Size | 180000 | 180000 | 180000 | 180000 | 180000 | 180000 |
| $v$ | 710000 | 710000 | 710000 | 710000 | 710000 | 710000 |
| $k$ | 512 | 512 | 512 | 512 | 512 | 512 |
| $D_{\text{emb}}$ | 512 | 512 | 1024 | 512 | 512 | 1024 |
| $D_{\text{up}}$ | 4096 | 4096 | 4096 | 4096 | 4096 | 4096 |
| $L^3$ Positions | 4 | 4, 16 | 4, 10, 16 | 4 | 4, 16 | 4, 16 |

*Table 6.* MoE Model Configurations

| Size | 800M 1 $L^3$ MoE Eq. | 800M 2 $L^3$ MoE Eq. | 1.5B 1 $L^3$ MoE Eq. | 1.5B 2 $L^3$ MoE Eq. |
|---|---|---|---|---|
| Decoder Layers | 20 | 20 | 24 | 24 |
| Hidden Size | 1024 | 1024 | 1536 | 1536 |
| Attention Heads | 32 | 32 | 32 | 32 |
| Attention Head Dim | 64 | 64 | 64 | 64 |
| Context Length | 2048 | 2048 | 2048 | 2048 |
| Batch Size (Tokens) | 131072 | 131072 | 131072 | 131072 |
| Peak Learning Rate | $3 \times 10^{-4}$ | $3 \times 10^{-4}$ | $3 \times 10^{-4}$ | $3 \times 10^{-4}$ |
| Min Learning Rate | $3 \times 10^{-5}$ | $3 \times 10^{-5}$ | $3 \times 10^{-5}$ | $3 \times 10^{-5}$ |
| Weight Decay | 0.1 | 0.1 | 0.1 | 0.1 |
| Gradient Clipping | 1 | 1 | 1 | 1 |
| Intermediate Dim | 1024 | 1024 | 1024 | 1536 |
| Tokens | $10^{10}$ | $10^{10}$ | $2 \times 10^{10}$ | $2 \times 10^{10}$ |
| Tokenizer Size | 180000 | 180000 | 180000 | 180000 |
| Experts Per Layer | 22 | 44 | 15 | 20 |
| Active Experts Per Layer | 4 | 4 | 6 | 4 |
| Load Balancing Weight | 0.01 | 0.01 | 0.01 | 0.01 |
| Router Z Loss Weight | 0.001 | 0.001 | 0.001 | 0.001 |

## A.3. Tokenizer Details

We used a standard BPE tokenizer with a vocab size of 180000 tokens. We trained this tokenizer from scratch on 10GB of data from the OLMO2 dataset in the SuperBPE codebase (OLMo et al., 2025; Liu et al., 2025).

## A.4. Implementation Details

### A.4.1. EMBEDDING ALLOCATION

We trained Algorithm 1 on 1GB of the same corpus as our tokenizer. Below is a Python implementation of Algorithm 1.

```python
def train_lzw(files, tok, k):

    lzw_counter = {}

    for s in range(tok.n_vocab):
        lzw_counter[(s,)] = 0

    for fn in files:
        f = open(fn).readlines()
        for line in tqdm.tqdm(f, mininterval=1):
            toks = tok.encode(line)
            last = 0
            cur = 1
            while cur < len(toks):
                while cur < len(toks) and tuple(toks[last:cur]) in lzw_counter:
                    cur += 1
                if cur > last+1:
                    lzw_counter[tuple(toks[last:cur-1])] += 1
                    lzw_counter[tuple(toks[last:cur])] = 1
                last = cur
                cur += 1

    lzw_counter = sorted(list(lzw_counter.items()), key=lambda x: x[1], reverse=True)
    alloc = [1 for _ in range(tok.n_vocab)]
    n_alloc = tok.n_vocab

    i = 0
    while n_alloc < target:
        if alloc[lzw_counter[i][0][-1]] < k:
            alloc[lzw_counter[i][0][-1]] += 1
            n_alloc += 1
        i += 1

    return alloc
```

### A.4.2. DOWNSTREAM EVALUATION

We use the standard GPTQ split of Wikitext2 to calculate perplexity with a context length of 2048 (Frantar et al., 2023; Merity et al., 2016). We use LM Eval 0.4.9.2 to run zeroshot evaluations and report "accuracy" numbers (not "acc_norm") (Gao et al., 2024). Since our models are "base models" and not SFT models, we do not use a chat template or any wrappers around the model during evaluation.

### A.4.3. INFERENCE SPEED

We measure inference speed with a single token and no KV cache and no sampling step. This bypasses overhead from orthogonal components to $L^3$ and represents the "worst-case" overhead $L^3$ might have during inference speed (these other components are not affected by $L^3$ and add overhead, reducing the actual proportion of time $L^3$ adds, if any). In our benchmark, we assume each token hits 512 contiguous embeddings with position sampled randomly from the embedding tables; this is strictly "harder" than an actual inference workload since not all tokens will hit the maximum allowed number of embeddings. To implement CPU offloading, we partition the model into 2 sections: before the first $L^3$ layer and after the first $L^3$ layer. Each section is captured in a single CUDA graph with `torch.compile(..., mode='max-autotune',`

`fullgraph=True)`. During the call for the first graph, *both* $L^3$ layers are moved to the GPU with a separate CUDA stream. After the first graph is called, the stream is synchronized and the second graph is called. Again, this is strictly harder than an actual inference workload since only the *first* $L^3$ layer needs to be moved to the GPU before it is hit — other layers can continue being moved after the first $L^3$ layer is hit.

### A.4.4. TRAINING

During training, we first sort the sequence and permute the incoming hidden states like in Figure 3. Then, we iterate over block diagonals. Below is a minimal PyTorch implementation of an $L^3$ layer. We also include an alternate forward pass using FlexAttention, which may be better suited for modern hardware. In all cases, we initialize $L^3$ layers using the standard linear layer initialization in Llama.

```python
def valid_collate_fn(batch):
    # Data collation function to produce inputs needed for L3 forward below
    # seqs are tokenized inputs
    # emb_alloc is the token ID for that embedding
    # bounds[i] is the start the region in emb_alloc corresponding to token i

    seqs = torch.stack([_[0] for _ in batch], dim=0)
    emb_alloc = batch[0][1]
    bounds = batch[0][2]

    seq_sort, fw = torch.sort(seqs.flatten())
    bw = torch.zeros(len(seq_sort), dtype=torch.int64)
    bw[fw] = torch.arange(len(seq_sort))

    unique, cts = torch.unique(seq_sort, return_counts=True)
    keep_cols = []
    starts = []
    ends = []
    for i in range(len(unique)):
        tidx = unique[i]
        tct = cts[i]
        starts += [len(keep_cols)]*tct
        keep_cols += list(range(bounds[tidx], bounds[tidx+1]))
        ends += [len(keep_cols)]*tct

    keep_cols = torch.tensor(keep_cols)
    starts = torch.tensor(starts)
    ends = torch.tensor(ends)

    return seqs, fw, bw, seq_sort, keep_cols, emb_alloc, starts, ends

class L3(torch.nn.Module):
    def __init__(self, h, n_emb, d_emb, d_up):
        super().__init__()

        self.d_emb = d_emb

        self.w_k = nn.Linear(h, n_emb, bias=False)
        self.w_v = nn.Linear(d_emb, n_emb, bias=False)
        self.w_mix = nn.Linear(d_up + h, h, bias=False)
        self.w_up = nn.Linear(d_emb, d_up, bias=False)

        self.norm_in = LlamaRMSNorm(h)
        self.norm_out = LlamaRMSNorm(h)

    @torch.compile(mode='max-autotune', fullgraph=True)
    def mask_logits_gemm_(self, A, B, C, seq_sort, last_token):
        score = A @ B.T
        score = torch.where(seq_sort.view(-1, 1) == last_token.view(1, -1), score, -float(
    'inf'))
```

```
        return score.softmax(dim=-1) @ C

    def mask_logits_gemm(self, A, B, C, seq_sort, last_token, starts, ends):
        b, t, d = A.shape
        out = torch.zeros(b, t, C.shape[-1], device=A.device, dtype=A.dtype)
        for i in range(b):
            start = starts[i]
            end = ends[i]
            out[i] = self.mask_logits_gemm_(A[i], B[start:end], C[start:end], seq_sort[i],
    last_token[start:end])
        return out

    def forward(self, input, fw, bw, seq_sort, keep_cols, emb_alloc, starts, ends, bb=512)
    :
        # bb is the size of the vertical dim of the block diagonal
        b, t, d = input.shape

        A = self.norm_in(input[fw])
        B = self.w_k.weight
        C = self.w_v.weight

        A = A.reshape(-1, d)[fw].reshape(-1, bb, d)
        B = B[keep_cols]
        C = C[keep_cols]
        emb_alloc = emb_alloc[keep_cols]

        seq_sort = seq_sort.reshape(-1, bb)
        starts = starts.reshape(-1, bb).min(dim=-1).values
        ends = ends.reshape(-1, bb).max(dim=-1).values

        comb_embs = self.mask_logits_gemm(A, B, C, seq_sort, emb_alloc, starts, ends)
        comb_embs = comb_embs.reshape(-1, self.d_emb)[bw].reshape(b, t, self.d_emb)
        comb_embs = self.w_up(comb_embs)
        return self.w_mix(torch.concat([self.norm_out(comb_embs), inputs], dim=-1))

    def forward_flexattn(self, input, fw, bw, seq_sort, keep_cols, emb_alloc):
        b, t, d = input.shape
        emb_alloc = emb_alloc[emb_alloc]

        def mask_fn(b, h, q_idx, kv_idx):
            return seq_sort[b, q_idx] == emb_alloc[kv_idx]

        mask = flex_attention.create_block_mask(
            mask_fn, B=b, H=None, Q_LEN=t, KV_LEN=emb_alloc.shape[0], _compile=True)

        A = self.norm_in(input[fw]).gather(
            1, fw.unsqueeze(-1).repeat(1, 1, d)).unsqueeze(1)
        B = self.w_k.weight.unsqueeze(0).unsqueeze(1)
        C = self.w_v.weight.unsqueeze(0).unsqueeze(1)

        comb_embs = torch.compile(flex_attention.flex_attention, mode='max-autotune')(
            A, B, C, block_mask=mask, enable_gqa=True)
        comb_embs = comb_embs.reshape(-1, self.d_emb)[bw].reshape(b, t, self.d_emb)
        comb_embs = self.w_up(comb_embs)
        return self.w_mix(torch.concat([self.norm_out(comb_embs), inputs], dim=-1))
```

### A.4.5. TUNED LENS

We follow the general philosophy of the tuned lens presented by Belrose et al. (2025). Specifically, we train a new layernorm and unembedding layer to minimize the KL divergence between a shortcutted model and the original model. That is, if $y = F_i(G_i(x))$ where $F$ is the rest of the model after decoder layer $i$ and $G_i$ is the prefix of the model up to and including

layer $i$, we train a function $S(z) = \text{Unembedding}(\text{LayerNorm}(z))$ to minimize $D_{\text{KL}}(S(G_i(x))\|y)$ for every $i$. We use a devset of 2048 sequences of sequence length 2048 and initialize $S$ to the layernorm and unembedding from the original model. We use AdamW with a weight decay of 0.01 and learning rate of $3 \times 10^{-4}$ with a batch size of 8192 tokens and 500 epochs and directly optimize the KL.

### A.5. FLOPs Counting

Here, we count the forward pass FLOPs required by a $\text{L}^3$ layer for a single token that activates $t$ embeddings. We ignore the cost of layernorms and concatenation since they are essentially free.

- $K_t x$: $2 \times t \times d_{\text{in}}$

- **Softmax**: $3 \times t$

- $V_t^T$ **Score**: $2 \times d_{emb} \times t$

- $W_{\textbf{up}}$ **(Attention Output)**: $2 \times d_{\text{up}} \times d_{\text{emb}}$

- $W_{\textbf{mix}}$ **(Concatenated Stream)**: $2 \times d_{\text{emb}} \times (d_{\text{emb}} + d_{\text{up}})$

Since we cap $t \leq k = 512$ in our experiments, the number of FLOPs added per token is very small. Empirically, we measure $\mathbb{E}[t] \approx 300$ for our standard experiment setup on FineWeb-Edu.

### A.6. Example Generations

Below ae three example generations of 256 tokens each from the 2.6B $L^3$ model with naive sampling with no temperature or top-$k$ limits.

---

**Prompt:** Four score and seven years ago

**2.6B 2 $L^3$ :** Four score and seven years ago our fathers brought forth, on this continent, a new nation, conceived in Liberty, and dedicated to the proposition that all men are created equal. Now we are engaged in a great civil war, testing whether that nation, or any nation so conceived and so dedicated, can long endure. We are met on a great battlefield of that war. We have come to dedicate a portion of that field, as a final resting place for those who here gave their lives that that nation might live. It is altogether fitting and proper that we should do this. But, in a larger sense, we cannot dedicate, we cannot consecrate, we cannot hallow this ground. The brave men, living and dead, who struggled here, have hallowed it, far above our poor power to add or detract. The world will little note, nor long remember what we say here, but it can never forget what they did here. It is for us the living, rather to be dedicated here to the great task remaining before us—that from these honored dead we take increased devotion to that cause for which they gave the last full measure of devotion—that we here highly resolve that these dead shall not have died in vain—that this nation, under God, shall have a new birth of freedom—and

---

**Prompt:** It is a truth universally acknowledged that

**2.6B 2 $L^3$ :** It is a truth universally acknowledged that if you give a monkey a long enough glass of water, it will learn how to deceive you. Excellent guys! Without further ado, let's take a close look at the ramifications of that experiment and your process of learning for good, or for the worse! Stated most simply, the monkeys below had been newly delivered from their mother's group and began developing their personalities among new monkey peers. In essence, every order begins with a "Significance Environment" which determines caste, lifetime social standing, and longevity. These monkeys continued to develop and compare new situations with known scenario sequences from before time to test order effects from childhood rearing experiences and into adulthood.

- The monkey order monkey in the figure left recieved a new human caretaker and was adapted to the general culture savvily, conditioned by novel stimuli.

- The counterposed monkey in the figure to the left recieved the exact same human caretaker but without this "form lesson", and was abandoned. Finally, he/she engaged in an environmental innovation and did not demonstrate changes in order.

- Initially, the Monkey #1 establishment and held unfulfilling rank and a decade of boredom. He/she spent 70 subcultures between five monkeys. Later he/she seemed to throw off that individuality and stereotyped behavior. We

**Prompt:** The vaunted protection that intellectually active adults get from Alzheimer's disease has a dark downside, a study released Wednesday has found

**2.6B 2 L$^3$ :** The vaunted protection that intellectually active adults get from Alzheimer's disease has a dark downside, a study released Wednesday has found. In a new study, scientists followed more than 16,000 adults for 20 years to assess the relationship between adult intellectual activity and rates of dementia. If engaged in intellectually stimulating activity for even just a short period of time, people reduced their risk of developing dementia by 45 percent. The risk for those who exercised intellectually for six or more months was reduced by 68 percent. There are dozens of other studies currently underway assessing relationships between, among other things, education, lack of alcohol consumption, IQ, and the development of Alzheimer's. The definition of intellectually active adults in this study was adults who did any of the following, per definition: read for leisure or at least 50 books or 100 newspapers or 200 magazines during the last 12 months, gave information to others about their professions, answered polls or studies, or held any other kind of office, teaching, or research job. They also had to have done any of the other measurements, such as taking courses, for social or personal interest or to increase their education. If they met none of these, they were classified as sedentary adults. We asked Tom Tierney in San Francisco if 60-somethings have the chops to think without a list of terms at their back to satisfy

The last prompt was taken directly from the FineWeb-Edu training set from a passage about Alzheimer's disease (ID `<urn:uuid:5d156165-181a-4195-a926-d51850c7b599>`). Prompting the *dense 2.6B model* with the same prompt and seed gives

**Prompt:** The vaunted protection that intellectually active adults get from Alzheimer's disease has a dark downside, a study released Wednesday has found

**2.6B Dense:** The vaunted protection that intellectually active adults get from Alzheimer's disease has a dark downside, a study released Wednesday has found. In a new study, scientists showed that adult neurogenesis helped protect young brains from damage caused by free radical exposure. Although adult neurogenesis is usually considered beneficial, its ability to ward off aging, memory loss and brain disease has been much debated. However, the new study published in Science and conducted by scientists at the National Institutes of Health's Buck Institute for Research on Aging, collects new evidence that might lead to new ways to maximize adult neurogenesis while minimizing its age-related risks. The study, led by David Holtzman, PhD, has been highlighted by two earlier papers. (Minko et al., Science, 2008) The British Antarctic Survey, BBC News A global warming of the Earth's temperature could cause civilization to collapse. In this report, written for the British Association for the Advancement of Science, Professor Philip Stott, Medieval Physicist says climate change is evident, human negotiations to halt that rise in global temperature are at a standstill, and the entire world will be burning by 2040. However, Professor Stott also seams this will not happen for at least another 30 years. Sustainable Development 2010: Complexity, Competitiveness and Institutions, Report published 2010 Professor Joel Kotkin is a graduate of Cornell University's School of International

which is subjectively of lower quality.

