# OpenReview forum: "$L^3$: Large Lookup Layers"
_ICML.cc/2026/Conference — ICML 2026 regular_

### Official Review · Reviewer_9xWU · 2026-03-06

**Soundness:** 3
**Presentation:** 2
**Significance:** 3
**Originality:** 3
**Overall Recommendation:** 4
**Confidence:** 3

**Summary:**

This paper extends the embedding layer for the sparsity of language models. Instead of preparing separated modules at the end of the model, this paper introduced sparsity utilizing the embedding layer (I guess this is not true, but it should be an unembedding layer, precisely), which is sparse by design.

**Compliance With Llm Reviewing Policy:**

Affirmed.

**Final Justification:**

The clarification during the discussion period improved my understanding, and I re-evaluate this paper in a positive light. While I understand it appropriately now, I was confused about the positioning of this paper only from the introduction and related work sections. I expect the updated version will include the clarification provided in the discussion.

**Key Questions For Authors:**

- I am not quite sure about the experimental setup. The first sentence of Section 4 says "L3 layers added between decoder layers". I guess that L3 is based on the unembedding layer, and it should be placed between the final decoder layer and the softmax. However, from the sentence here, I am confused whether L3 could be placed between each decoder layer. Where is L3 placed among the entire architecture?

**Limitations:**

- Authors acknowledged the possibility of using both L3 and MoE, but such experimental setup is not included in the current paper.

**Strengths And Weaknesses:**

# Strength
- It is interesting to extend an (un)embedding layer for the sparsity.
  - By the way, I feel it is very confusing to mention an unembedding layer as "tokenization embedding table" in the introduction. I first got an impression that the propoese method utilizes the first embedding layer, while it actually uses the so-called unembedding layer in the decoder part.
- Incorporating L3 improves the perplexity and downstream performance compared to a vanilla architecture.

# Weakness
- If my understanding is correct, the proposed method is not "sparse" as defined in Section 2. Since the proposed L3 layer employs an attention mechanism, the final obtained hidden state should be a mixture of all parameters. In this sense, I consider the proposed method just adds active parameters to each layer, and it simply contributes to the performance improvement. Even if the proposed method truly "lookup" one single embedding from V_t, it still uses all parameters to calculate the attention scores, and they should be counted as "active". Therefore, I do not consider that the proposed method really works to increase the sparseness as the authors intend.
- The proposed method is not clearly explained. I understand the L3 module, thanks to Section 3, but I am not quite sure how we should use this module in the architecture. Considering the nature of the idea to use the (un)embedding layer, I expected that L3 is placed at the end of the entire architecture. But Section 4 suggests that one can replace MLP with L3, which is confusing.
- Also, the writing can be improved. Many figures are not mentioned around the figure itself. For example, Figure 1 is first mentioned at the end of p5. Figures are also sufficiently explained (e.g., what is dual and triple in Figure 6?).
- The proposed method is not compared with MoE on the downstream task. I understand the proposed method does not focus on the replacement of MLP, and it can be used with MoE, but I think it is not a valid reason to exclude the comparison with MoE. If the proposed method is truly sparse (as I mentioned above), it should be compared with other methods for the sparsity on the downstream tasks, in addition to perplexity.
- Although extending the embedding layer is interesting, I feel this idea seems rephrase of existing research on NLP that parepare multiple embeddings for multi-sense words, such as https://aclanthology.org/2021.eacl-main.45 .

---

> ### Author Rebuttal · Authors · 2026-03-30
>
> Dear reviewer, thank you for your review. **Due to the 5000 character limit, we did not include the same header we included in our rebuttals for other reviewers. However, we request that you to read it since it applies to your review too.**
>
> We think you may have fundamentally misunderstood the paper and how L3 works. In a L3 layer, there are two large embedding matrices $V$ and $K$ (or one if you weight-tie them together, but that is an implementation detail). $V_t$ and $K_t$ in Figure 2 refer to the **subset** of $V$ and $K$ that corresponds to a token ID $t$. Critically, $t$ is the **input token ID**, not the output of some modeling process in the current autoregressive step. This is why we can do things like offloading, since the active L3 parameters are known the moment t is generated. L3 is sparse because we are using a subset of $V$ and $K$ to compute attention over. The effective sparsity mask is just the rows of $V$ and $K$ that are active for a given $t$. This is why $V$ and $K$ are embedding tables, and not **un**embedding tables like your review suggests. In fact, if you consider the case where each token ID $t$ gets only one embedding in $K$ and $V$, then this is just the regular embedding table at the beginning of a model.
>
> The key idea in L3 is that $V_t$ and $K_t$ are fixed with respect to $t$ so that the sparsity mask is static and not dependent on the context. This admits systems benefits such as offloaded inference and also faster convergence due to not having to learn a discrete routing. **Within** $V_t$ and $K_t$, the embeddings are aggregated via attention with the hidden state $x$, which does contain context information since we are guaranteed that at least one attention layer has happened in the past (if this is not true, then again this is just a regular old token embedding table). In our experiments, we placed L3 layers between decoder layers. The premise of your question on the experimental setup seems to be based on viewing L3 as an unembedding layer, which it is not. We hope that our explanation above clears things up.
>
> ## Specific Questions:
>
> - If my understanding is correct ...
>   - See above
> - The proposed method is not clearly ...
>   - L3 is not intended to be a replacement for any of the usual layers found in a transformer architecture. It is an embedding layer intended to be placed in various parts of the model between decoder layers.
> - Also, the writing ...
>   - Dual and triple refer to using 2 and 3 L3 layers. We will attempt to reorder the figures to be closer to their references.
> - The proposed method is not compared ...
>   - Below are the zeroshot numbers for the MoE equivalent models. We will add these to an updated version of the manuscript. All MoEs underperform their equivalent L3s in Table 1.
>
> |         Model        |  Avg  |  ArcC |  ArcE | HSwag |  PiQA | WinoG |
> |:--------------------:|:-----:|:-----:|:-----:|:-----:|:-----:|:-----:|
> | 800M 1 L3 MoE equiv. | 48.18 | 27.25 | 60.27 | 34.79 | 67.36 | 51.22 |
> | 800M 2 L3 MoE equiv. | 48.59 | 27.60 | 60.48 | 35.28 | 68.39 | 51.22 |
> | 1.5B 1 L3 MoE equiv. | 51.68 | 30.63 | 65.65 | 38.89 | 70.18 | 53.04 |
> | 1.5B 2 L3 MoE equiv. | 52.83 | 31.72 | 66.54 | 41.23 | 70.52 | 54.13 |
>
> - Although extending the embedding ...
>   - Our understanding of PolyLM is that it aims to learn which meaning of a token is being used in an input sequence (i.e. “Word Sense Induction”) via a series of regularization losses (S3.5 in PolyLM). PolyLM has three components, an “input network” that takes a token level embedding, a “disambiguation network” that outputs an embedding vector per token, and a “prediction network” that makes a prediction over sememes. In between the disambiguation and prediction networks, the output embedding of the disambiguation network attends to the sememe embeddings, which are shared over the final sememe unembedding layer and input token embedding layer.
>   - This is different from the goal, architecture, and training setup of L3. In L3, our goal is to achieve systems-friendly structured sparsity in large-scale autoregressive next-token modeling. While both architectures use an attention mechanism, the fundamental goal, implementation, and scale of the two architectures is different. In L3, we investigated how to allocate embeddings to token IDs through our information-theoretic LZW algorithm.  There was also a large systems co-design component of L3, where we showed how to efficiently train and perform fast offloaded inference with L3, which is critical for deployment at the LLM scale.
> - I am not quite sure ...
>   - See the large response above

---

> > ### Author Rebuttal · Reviewer_9xWU · 2026-04-03
> >
> > Thank you very much for the clarification.
> > I consider my previous use of the term "unembedding" was inappropriate, and my statement "it uses all parameters to calculate attention scores" was also incorrect. I appreciate the clarification that L3 operates on a subset of K and V determined by the token ID.
> > Also, thank you for the additional experimental results, which should be helpful for readers. I will raise the score accordingly by the end of the discussion period.
> >
> > Let me clarify my original question, which was more about the nature of sparsity in L3.
> > While L3 indeed selects a subset of parameters via static routing, the computation within that subset is still dense because of the attention-based aggregation.
> > In this sense, L3 appears to perform dense computation over a sparse subset of parameters.
> >
> > Regarding your response:
> >
> > > if you consider the case where each token ID $t$ gets only one embedding in $K$ and $V$, then this is just the regular embedding table at the beginning of a model.
> >
> > I agree with this point. In that extreme case, the computation is clearly sparse because they do not have to use an attention mechanism.
> > However, in the general case where multiple embeddings are assigned to each token and combined with attention, I consider that the behavior differs from standard sparse routing, where only a small number of experts are activated without dense mixing.
> >
> > Therefore, my concern is not about whether a subset is selected, but rather whether this form of sparsity is conceptually comparable to other sparse architectures (MoE). In particular, it would be helpful if the paper could clarify how this notion of sparsity relates to existing definitions in detail.
> >
> >
> > In addition, since sparsity is one of the central motivations of the paper, I think it would strengthen this paper to further clarify in what sense L3 should be considered “sparse.”
> > As mentioned above, while the method activates only a subset of parameters, the computation within that subset remains dense, which seems different in nature from commonly studied sparse architectures to my understanding.
> > Clarifying this point would help better position the contribution.
> >
> > Furthermore, while comparisons to MoE are useful, I find that L3 appears conceptually closer to embedding expansion or memory-augmented approaches. In this sense, it would be helpful to include comparisons with such approaches.

---

> > > ### Author Response · Authors · 2026-04-05
> > >
> > > > my concern is not about whether a subset is selected, but rather whether this form of sparsity is conceptually comparable to other sparse architectures (MoE)...
> > > > the computation within that subset remains dense, which seems different in nature from commonly studied sparse architectures to my understanding
> > >
> > > We believe there may be some confusion between the classical sense of sparsity (e.g. a sparse matrix in linear algebra) and what we are referring to as "structured sparsity" in our discussion in Section 2. Your statement that "the method activates only a subset of parameters, the computation within that subset remains dense" is correct **but also applies to MoEs**, the most commonly studied sparse architecture. In an MoE, for a single token, a subset of the experts is chosen to process the token. **Each expert has the same architecture as a dense MLP block**, which means that **the computation for a given expert is dense as you described.** For example, the HF implementation of GPT-OSS (an open source MoE) clearly shows that each expert is actually doing dense computation on its input tokens: https://github.com/huggingface/transformers/blob/main/src/transformers/models/gpt_oss/modeling_gpt_oss.py#L100.
> > >
> > > The reason why dense operations are preferred in modern structured sparse architectures is because they can be implemented efficiently on hardware. As long as the parameter sparsity is "separated out" from the computation, we are still effectively getting parameter sparsity without the hardware overhead of arbitrary sparsity patterns. In contrast, classical sparsity is limited to n:m (e.g. 2:4) sparsity for hardware acceleration. This makes it difficult to realize both quality and throughput gains, which is why n:m has largely fallen out of favor vs MoEs.
> > >
> > > > Furthermore, while comparisons to MoE are useful, I find that L3 appears conceptually closer to embedding expansion or memory-augmented approaches. In this sense, it would be helpful to include comparisons with such approaches.
> > >
> > > L3 is positioned as another way to scale sparsity within models, not as a replacement for the MoE architecture. For example, the sparsity rate of a single L3 layer is much higher than that of a MoE layer (>500X vs ~30X). We believe that both have their place in modern models. The purpose of our experiments with MoEs was to show that L3 has similar scaling properties as MoEs.
> > >
> > > In Section 2.3 (our Background section), we compared against other embedding based approaches. The conceptually closest work to ours is Engrams, which is concurrent work since it came out *one week* before we finished our paper. Engrams places embedding tables throughout the model and uses local suffix information to select a set of embeddings per token. Similar to L3, these embeddings are aggregated based on the current hidden state with full context information. However, unlike L3, Engrams contains many additional steps such as n-gram hashing, tokenizer compression, and pooling. As we wrote in our response to D4K3, we leave a full comparison to Engrams for the future. Their training code has not been fully released yet, making it practically difficult to compare L3 and Engrams right now. We also explained to D4K3 why we did not compare against Product Key Networks (PKNs) such as UltraMem. PKNs select which embeddings to use based on the hidden state of the current token, which means that PKNs perform context-dependent routing. This means that they do not possess the systems benefits of L3 such as parameter offloading.
> > >
> > > One experiment we would like to highlight is Figure 7C, where we compared our LZW allocation algorithm to a uniform allocation algorithm. Google's recent Gemma 4 series of model uses "per layer embeddings" (PLEs), which are large embedding banks that allocate a single embedding per token. Like L3, PLEs perform context-independent routing, meaning that they can be offloaded. However, since there is only a single embedding per token, PLEs do not perform context dependent aggregation. PLEs can be seen as a strictly weaker version of the uniform allocation case in Figure 7C, which itself is inferior to the LZW allocation used in L3.
> > >
> > > Due to logistical issues, we do not currently have access to enough compute to run more training experiments before the end of the discussion period. However, we agree that it would be very interesting to compare different embedding architectures in an isolated setting, which none of the other papers we mentioned above current do either. Our hypothesis is that the main component that makes embedding table architectures work is the ability for the model to store information directly in parameters like a lookup table. This is why we think that a simple implementation of this idea that prioritizes systems co-design like L3 works so well. There are a few other components of this that seem to be important, such as how embeddings are allocated to tokens, which we studied with our novel LZW-based algorithm.

---

### Official Review · Reviewer_D4K3 · 2026-03-13

**Soundness:** 3
**Presentation:** 3
**Significance:** 3
**Originality:** 3
**Overall Recommendation:** 4
**Confidence:** 5

**Summary:**

This paper introduces the Large Lookup Layer (L3), an architecture for building sparse language models. Inspired by the sparse and static routing of tokenizer embedding tables, L3 generalizes this concept to the function within transformer decoder layers. The core idea is to equip the model with large and learned embedding tables. For a given input token, L3 uses static and token-based routing​ to select a fixed set of embeddings. The model's current hidden state then performs a context-dependent aggregation (via an attention-like mechanism) over these selected embeddings, allowing it to look up and incorporate cached information. The authors empirically validate L3 by training Llama-based models with up to 2.6B active parameters. The results show that L3 models outperform iso-compute dense baselines and iso-parse MoE models on both language modeling perplexity and a range of downstream NLP tasks, while maintaining high training and inference efficiency. Overall, L3 is a highly efficient architecture that is well-suited for scaling law.

**Compliance With Llm Reviewing Policy:**

Affirmed.

**Key Questions For Authors:**

Please refer to 'Weaknesses'.

**Limitations:**

Please refer to 'Weaknesses'.

**Strengths And Weaknesses:**

Strengths: The paper is technically sound. The paper is also clearly written. The core claim, the L3 architecture, improves model performance over dense and MoE baselines. The experiments are also conducted with multiple model sizes (800M, 1.5B, 2.6B active params), and detailed ablation studies are provided, showing the effectiveness and efficiency of the L3 architecture. By offering a sparse architecture with static routing, L3 directly addresses the system bottlenecks (e.g., difficulty with offloading, load-balancing issues) associated with mainstream sparse MoE-based models. The demonstrated ability to offload parameters to the CPU with negligible inference overhead is a significant practical advantage for deployment.

Weaknesses: While the paper correctly cites the concurrent research about Engrams, a more direct, quantitative comparison with Engrams (and other studies such as UltraMem) is still necessary. These experiments will highlight the advantages or disadvantages of the simpler L3 method compared to the more complex mechanisms of Engrams, thus more clearly demonstrating the innovation of L3. Another problem is that the MoE baseline is too simple. The current MOE architecture is achieved through sophisticated expert partitioning and complex load-balancing mechanisms. How to design the MOE baseline is also a crucial problem.

---

> ### Author Rebuttal · Authors · 2026-03-30
>
> Dear reviewer, thank you for your review. As noted by you and other reviewers, the key proposal in our paper, the L3 sparse architecture, is a “technically sound,” “clever[,] and elegant idea that is strongly motivated” (D4K3, FnRu). The key idea in L3 is to use token-specific routing to a lookup-based embedding table to achieve sparsity while using context-based embedding aggregation to preserve quality. This allows L3 to “directly [address] the systems bottlenecks … associated with mainstream sparse MoE-based models,” (D4K3) such as through minimal overhead offloading (D4K3, kBqv), which is a “significant practical advantage for deployment” (D4K3).
>
> Our experiments across multiple model sizes and architectures “strongly support [our conclusions]” (FnRu), such as that “L3 follows a strong scaling law and can be competitive with previous MoE-based models at similar model sizes” (kBqv). We are glad that reviewers found our paper to be “well [and] clearly written” (FnRu, D4K3) and appreciated our “detailed ablation studies [that show] the effectiveness and efficiency of the L3 architecture” (D4K3), such as its ability to “[improve] perplexity and downstream performance [over] a vanilla [model]” (9xWU).
>
> Below, we have responded to each of your questions separately. Due to logistical issues, we no longer have access to the compute we ran our original training experiments on. Therefore, we were unable to run additional training experiments during the rebuttal period. We were, however, able to answer other questions (such as timing numbers and downstream evals) that required fewer resources.
>
>
>
> - While the paper correctly cites the concurrent research about Engrams, a more direct, quantitative comparison with Engrams (and other studies such as UltraMem) is still necessary. These experiments will highlight the advantages or disadvantages of the simpler L3 method compared to the more complex mechanisms of Engrams, thus more clearly demonstrating the innovation of L3. Another problem is that the MoE baseline is too simple. The current MOE architecture is achieved through sophisticated expert partitioning and complex load-balancing mechanisms. How to design the MOE baseline is also a crucial problem.
>   - Engrams, UltraMem, and Product Key Networks:
>     - We agree that it would be a very interesting problem to study how various “embedding table” architectures compare to each other. Unfortunately, we do not have the resources to perform this study right now and therefore leave it for future work. Furthermore, since Engrams’s training kernels have not been released yet, running experiments with Engrams is currently practically difficult. One key difference that separates our work and Engrams from UltraMem and PKNs in general is that PKNs select keys based on a query vector formed from the current hidden state. This means that they effectively perform context-based routing like an MoE, which makes the systems benefits of L3 like offloading impossible to realize for them.
>     - Our hypothesis is that the main component that makes embedding table architectures work is the ability for the model to store information directly in parameters like a lookup table. This is why we think that a simple implementation of this idea that prioritizes systems co-design like L3 works so well. There are a few other components of this that seem to be important, such as how embeddings are allocated to tokens, which we studied with our novel LZW-based algorithm.
>   - MoE baseline: Our MoE baselines were trained with the guiding principles in OlMoE, which include relatively modern techniques (router z loss, load balancing, etc.). Since L3 is not intended to be a replacement for MoEs, the main purpose of the MoE baseline was to see how well L3 scaled with sparsity. We fully agree that the MoE architecture can be optimized in many different ways (e.g. shared experts), but the core sparsity scaling behavior of a MoE is largely consistent modulo these choices, so we think our MoE baseline is reasonable.

---

### Official Review · Reviewer_kBqv · 2026-03-13

**Soundness:** 3
**Presentation:** 2
**Significance:** 3
**Originality:** 3
**Overall Recommendation:** 4
**Confidence:** 4

**Summary:**

This paper proposes the Large Lookup Layer, a new sparse architecture for language models that generalizes embedding tables to transformer decoder layers. Instead of dynamic routing like in Mixture-of-Experts (MoE), L3 uses static token-based lookup routing, retrieving learned embeddings for each token to perform computation in a more hardware-friendly way.

The method includes a systems-efficient layer design and an information-theoretic algorithm for allocating embeddings across tokens. Experiments with models up to 2.6B active parameters show that it outperforms dense models and similarly sparse MoE models on language modeling and downstream tasks.

**Compliance With Llm Reviewing Policy:**

Affirmed.

**Key Questions For Authors:**

Please see the comments above for more details.

**Limitations:**

It seems that L3 works well at a relatively small scale. If possible, I would be happy to see results at larger scales, such as 14B, 35B, or 100B models.

**Strengths And Weaknesses:**

Strengths

1. The paper proposes a new architecture that is compatible with existing MoE and self-attention designs by introducing a lookup-table–based embedding layer, which potentially enlarges the scaling space for LLMs.

2. It introduces a GPU-efficient scheduling strategy that attempts to overlap the large memory transfer between CPU and GPU.

3. Experiments show that L3 follows a strong scaling law and can be competitive with previous MoE-based models at similar model sizes.

Weaknesses

1. curious about the plan for enabling reasoning capabilities (e.g., CoT stages) with the L3 architecture.

2. It would be helpful if the authors could further describe how L3 behaves during chunked prefill and decoding. In particular, more discussion on the efficiency benefits in both prefilling and decoding stages, especially in serving scenarios, would strengthen the paper.

---

> ### Author Rebuttal · Authors · 2026-03-30
>
> Dear reviewer, thank you for your review. As noted by you and other reviewers, the key proposal in our paper, the L3 sparse architecture, is a “technically sound,” “clever[,] and elegant idea that is strongly motivated” (D4K3, FnRu). The key idea in L3 is to use token-specific routing to a lookup-based embedding table to achieve sparsity while using context-based embedding aggregation to preserve quality. This allows L3 to “directly [address] the systems bottlenecks … associated with mainstream sparse MoE-based models,” (D4K3) such as through minimal overhead offloading (D4K3, kBqv), which is a “significant practical advantage for deployment” (D4K3).
>
> Our experiments across multiple model sizes and architectures “strongly support [our conclusions]” (FnRu), such as that “L3 follows a strong scaling law and can be competitive with previous MoE-based models at similar model sizes” (kBqv). We are glad that reviewers found our paper to be “well [and] clearly written” (FnRu, D4K3) and appreciated our “detailed ablation studies [that show] the effectiveness and efficiency of the L3 architecture” (D4K3), such as its ability to “[improve] perplexity and downstream performance [over] a vanilla [model]” (9xWU).
>
> Below, we have responded to each of your questions separately. Due to logistical issues, we no longer have access to the compute we ran our original training experiments on. Therefore, we were unable to run additional training experiments during the rebuttal period. We were, however, able to answer other questions (such as timing numbers and downstream evals) that required fewer resources.
>
> - curious about the plan for enabling reasoning capabilities (e.g., CoT stages) with the L3 architecture.
>
>   - We think it would be very interesting to study how L3 interacts with reasoning capabilities. For example, based on your comment on chain of thought stages, perhaps a model could learn to cache chains of thoughts in the embedding tables. Our experiments in the paper focused on pretraining since we viewed L3 as an architectural change. In our construction, L3 layers are static during inference. However, it is very possible that something like test time training or context compression could take advantage of L3 by dynamically updating the embeddings (perhaps with CoTs). One possibility that we have thought of and are leaving for future work is to do finetuning with L3, where L3 layers are treated like adapters for “sparse expansion” of a model.
>
> - It would be helpful if the authors could further describe how L3 behaves during chunked prefill and decoding. In particular, more discussion on the efficiency benefits in both prefilling and decoding stages, especially in serving scenarios, would strengthen the paper.
>   - Since L3 does not perform sequence mixing, chunked prefill isn’t really different from regular prefill, which is just the training setting described in section 3.2. In terms of whether offloading behaves differently, there are two regimes in the “roofline analysis” of how L3 behaves when run with a batch size greater than 1. First, in the compute-bound case, the cost of computation per decoder layer masks the cost of prefetching L3.
>   - In the memory bound setting (batch size <300 on a datacenter GPU) the amount of time needed to process a single decoder layer is bounded by the time needed to load its weights from HBM. Based on what we observed on L257, we would expect to load 3-4M L3 parameters for our 2.6B model and a batch size of 300 (i.e. the worst case scenario for offloading). PCIE5 has a bandwidth of 64GB/s per direction, which means loading these parameters in BF16 should take around 1e-4s. A single decoder layer in that same model has 68M parameters, which should take 2e-5s to load from HBM. Even in the worst memory-bound case, only about 20% of the decoder layers should be sufficient to completely mask the cost of offloading, which aligns with the batch size 1 case in Table 2. If we used weight tying or quantization to further reduce the size of L3, this would make offloading even easier to mask. We ran some additional timing numbers for FnRu that validates these back-of-the-envelope calculations, which we included in our response to them.
>
> - It seems that L3 works well at a relatively small scale. If possible, I would be happy to see results at larger scales, such as 14B, 35B, or 100B models.
>   - Unfortunately, this is not something that we can afford to do right now. Training a 100B total parameter model for even 100B tokens would cost well over $30K and training a 100B active parameter model would cost a few million dollars. We agree that it would be interesting to scale L3 up and leave that for future work.

---

> > ### Author Rebuttal · Reviewer_kBqv · 2026-04-01
> >
> > Thank authors for clarifying my concerns and proposing the possible methods for CoT. I will keep my score.

---

### Official Review · Reviewer_FnRu · 2026-03-17

**Soundness:** 3
**Presentation:** 4
**Significance:** 3
**Originality:** 4
**Overall Recommendation:** 5
**Confidence:** 4

**Summary:**

This paper introduces a new modification to the standard transformer architecture by adding large, sparse matrices. These “L3” layers generalize the idea of an embedding table to middle layers in a way that allows for fast training and inference. Specifically, L3 uses static context-independent routing to select a fixed set of embeddings per token ID (which also enables prefetching), and acts as a cache that allows a model to “shortcut computation” around decoder layers. The authors run experiments on Llama 3-style models with parameters ranging from 800M to 2.6B, and compare dense, MoE and L3 architectures with similar total FLOPs (isoFLOPs).

**Compliance With Llm Reviewing Policy:**

Affirmed.

**Key Questions For Authors:**

* Figure 6: the loss curves look strangely periodic - is this due to strange batching? What is your training infra used to generate these curves?
* Table 2: What does First L3 layer 4 mean? Does it mean that you insert the first of 2 L3 layers at decoder layer 4, which gives the system enough time to do offloading?

**Limitations:**

Yes

**Strengths And Weaknesses:**

## Strengths

The idea of generalizing embeddings to intermediate layers is a clever and elegant idea that is strongly motivated. They run a handful of experiments across multiple model sizes (800M - 2.6B dense models) and architectures (corresponding MoE and L3 models) that strongly support their conclusions. The paper is well written, and novel enough that it will likely generate significant interest from the research community.

## Weaknesses

* Figure 1: The figure is lacking results for the iso-sparse MoE at 800M (blue line) ~5B params point, as well as the 2.6B (green) ~7B params point. Including these values would increase confidence in the results
* The results relating to inference with L3 are limited. For example, it would be helpful to include values similar to Table 2 for batch size larger than 1
* As with all architecture papers, I am slightly skeptical of results that don’t do hyperparameter sweeps. We know there is a relationship between model parameters and learning rate etc. for dense and MoE models. However, I understand that such hyperparameter sweeps might be costly/prohibitive (esp. with pretraining).

Minor
* Figure 1: The caption should make it clear that “+” is L3 layer
* Figure 1: It would be helpful to display the same data with FLOPs on the x-axis just as a way to help visualize the results (maybe in the appendix) - presumably these are isoFLOPs, so they would be overlapping on the x-axis? I assume this data is selected from the perplexity plots in Figure 6?
* Table 1: Would be helpful to bold and underline best and second-best values
* Line 263: It would be helpful to expand on why L3 would be compatible with context parallelism. Have the authors explored this?
* Figure 8: What exactly is meant by L3 MoE equivalent?
* Figure 10: It would be helpful to mention in the caption that Layers 4 and 16 have L3 modules
* Line 404: Typo “This suggests that weight tying is strictly and that…”

---

> ### Author Rebuttal · Authors · 2026-03-30
>
> Dear reviewer, thank you for your review. As noted by you and other reviewers, the key proposal in our paper, the L3 sparse architecture, is a “technically sound,” “clever[,] and elegant idea that is strongly motivated” (D4K3, FnRu). The key idea in L3 is to use token-specific routing to a lookup-based embedding table to achieve sparsity while using context-based embedding aggregation to preserve quality. This allows L3 to “directly [address] the systems bottlenecks … associated with mainstream sparse MoE-based models,” (D4K3) such as through minimal overhead offloading (D4K3, kBqv), which is a “significant practical advantage for deployment” (D4K3).
>
> Our experiments across multiple model sizes and architectures “strongly support [our conclusions]” (FnRu), such as that “L3 follows a strong scaling law and can be competitive with previous MoE-based models at similar model sizes” (kBqv). We are glad that reviewers found our paper to be “well [and] clearly written” (FnRu, D4K3) and appreciated our “detailed ablation studies [that show] the effectiveness and efficiency of the L3 architecture” (D4K3), such as its ability to “[improve] perplexity and downstream performance [over] a vanilla [model]” (9xWU).
>
> Below, we have responded to each of your questions separately. Due to logistical issues, we no longer have access to the compute we ran our original training experiments on. Therefore, we were unable to run additional training experiments during the rebuttal period. We were, however, able to answer other questions (such as timing numbers and downstream evals) that required fewer resources.
>
>
> - Figure 1: The figure is lacking results for ...
>   - We will try to run these experiments for a camera ready version of the paper. However, we do not currently have access to enough compute for this and renting dedicated cloud compute would cost at least a thousand dollars.
> - The results relating to inference with L3 ...
>   - Below are numbers for bs=8 and bs=300 (the memory bound “limit” in most roofline analyses). We could not find an A100 (the GPU used in the paper) in the shared cluster we have access to, but were able to borrow a single B200 to run some timing numbers on. For reference, we have rerun the bs=1 numbers from the paper on the B200 as well. The relationship between offloading and inference overhead is consistent across batch size sizes in the memory bound regime, which is harder to optimize for than the compute bound regime.
>
> |   Dense Model  |       776       |       492       |        487        |
> |:--------------:|:---------------:|:---------------:|:-----------------:|
> | First L3 Layer | bs=1 toks/s/seq | bs=8 toks/s/seq | bs=300 toks/s/seq |
> |        1       |       692       |       424       |        396        |
> |        2       |       711       |       440       |        421        |
> |        3       |       743       |       459       |        450        |
> |        4       |       768       |       476       |        468        |
> |        5       |       772       |       487       |        484        |
>
> - As with all architecture papers ...
>   - We agree that it would be beneficial to sweep hyperparameters and study optimal scaling laws for L3. We view this as future work (and it is also outside of our compute budget). Based on our experiments, it is clear to us that L3 works as an architecture, which is often the first step towards studying its scaling behavior.
>
> - Figure 1: It would be helpful to display ...
>   - Correct, this plot was generated from Table 1. We will add this plot.
> - Line 263: It would be helpful to expand on why L3 would be compatible with context parallelism. Have the authors explored this?
>   - We have not personally implemented context parallelism with L3. The statement on L263 was just saying that due to the ability to offload L3 parameters, context parallelism with L3 should be easier to implement than context parallelism with an MoE, where you would have to contend with expert sharding or high memory pressure due to needing all experts to live on device memory.
> - Figure 8: What exactly is meant by L3 MoE equivalent?
>   - An iso-FLOP and iso-sparse MoE model. We will clarify this in the paper.
> - Typos
>   - We will fix these in an updated version of the paper.
> - Figure 6: the loss curves look ...
>   - We used an EMA of 0.9 when generating the plots, which may have contributed to it looking cyclic. We used a shuffle buffer of a fixed size when streaming Fineweb from Huggingface, which may have also contributed to this. Our training setup is based off of Huggingface’s transformers with torch native tooling (eg FSDP). Since all of our plots for all models exhibit the same behavior, we are fairly certain that this is due to how we streamed the dataset.
> - Table 2: What does First L3 layer 4 mean? ...
>   - Yes, that is correct.

---

> > ### Author Rebuttal · Reviewer_FnRu · 2026-04-06
> >
> > Thanks for the detailed rebuttal. I will keep my score

---

### Decision · Program_Chairs · 2026-04-30

**Decision:**

Accept (regular)

**Comment:**

This submission proposes Large Lookup Layers (L3), a sparse architecture that extends embedding-style lookup to transformer decoder layers through static token-based routing and context-dependent aggregation. The paper makes a convincing empirical case that this design improves over comparable dense and sparse baselines while retaining favorable systems properties.

The main remaining concerns are about positioning and completeness rather than the core claim. The paper would benefit from a clearer discussion of how its notion of sparsity relates to standard sparse architectures, as well as a more direct comparison to concurrent embedding- or memory-based approaches such as Engrams and related methods. Some aspects of the serving evaluation could also be presented more fully in the paper itself rather than in the rebuttal. Overall, however, the central contribution is novel, the experimental support is broad for the scale studied, and the paper makes a credible case that L3 provides a useful alternative axis of sparse scaling. A final version that sharpens the positioning and incorporates the main clarifications from the discussion would be stronger.